# Effects of *Epichloë* *sinensis* Endophyte and Host Ecotype on Physiology of *Festuca sinensis* under Different Soil Moisture Conditions

**DOI:** 10.3390/plants10081649

**Published:** 2021-08-11

**Authors:** Wenbo Xu, Miaomiao Li, Weihu Lin, Zhibiao Nan, Pei Tian

**Affiliations:** 1State Key Laboratory of Grassland Agro-Ecosystems, Key Laboratory of Grassland Livestock Industry Innovation, Ministry of Agriculture and Rural Affairs, College of Pastoral Agriculture Science and Technology, Lanzhou University, Lanzhou 730020, China; xuwb16@lzu.edu.cn (W.X.); limm16@lzu.edu.cn (M.L.); linwh15@lzu.edu.cn (W.L.); zhibiao@lzu.edu.cn (Z.N.); 2Institute of Rural Development, Gansu Provincial Academy of Social Sciences, Lanzhou 730071, China

**Keywords:** *Festuca sinensis*, *Epichloë* *sinensis*, host ecotype, water deficiency, growth, physiology

## Abstract

This study explored the effects of the *Epichloë* *sinensis* endophyte on growth, photosynthesis, ionic content (K^+^ and Ca^2+^), phytohormones (abscisic acid—ABA, cytokinin—CTK, indolE−3-acetic acid—IAA, and gibberellin—GA), and elements—C, N, P (in the shoot and root) in two ecotypes of *Festuca sinensis* (ecotypes 111 and 141) under different soil water conditions (35% and 65% relative saturation moisture content (RSMC)). The results showed that 35% RSMC inhibited the plants’ growth, and compared with 65% RSMC, there was a significant (*p* < 0.05) decrease in the growth and photosynthesis indices, the contents of CTK and GA, Ca^2+^ concentration, and the contents of C, N, and P (in both the aboveground and underground parts) under 35% RSMC. *E.* *sinensis* had beneficial effects on host growth and stress tolerance. Under both 35% and 65% RSMC, the presence of *E.* *sinensis* significantly (*p* < 0.05) increased host plant height, tiller number, root length, root volume, shoot dry weight, chlorophyll content, and the rate of photosynthesis of both ecotypes. Furthermore, the shoot C, N, and P contents in plants infected with *E.* *sinensis* (E+) from the two ecotypes, under both conditions of RSMC, were significantly (*p* < 0.05) higher than those in corresponding plants that were not infected with *E.* *sinensis* (E−). Under 35% RSMC, the contents of ABA, K^+^, Ca^2+^, and root P contents in E+ plants were significantly (*p* < 0.05) higher than those in corresponding E− plants in both ecotypes. However, under 65% RSMC, root C, N, and P contents in E+ plants of ecotype 111 and 141 were significantly (*p* < 0.05) higher than those in corresponding E− plants. In addition, the host ecotype also had effects on host growth and stress tolerance; the growth and photosynthetic indices of ecotype 141 were significantly (*p* < 0.05) higher than those of ecotype 111 under 35% RSMC, which suggested that ecotype 141 is more competitive than ecotype 111 under water deficiency conditions. These findings suggest that the endophyte improved the host plant resistance to water deficiency by maintaining the growth of the plant, improving photosynthesis, accumulating K^+^ and Ca^2+^, promoting nutrient absorption, and adjusting the metabolism of plant hormones.

## 1. Introduction

Drought is one of the major environmental factors determining plant productivity and distribution [1]. This condition is rapidly increasing on a global scale. Furthermore, along with salinity, the average yields for most major crop plants declines by more than 50% [2]. The water deficit in the soil and high temperatures are the two important factors that cause drought stress in temperate environments, which results in the inhibition of plant photosynthesis, ion imbalance, and metabolic disorder, ultimately affecting plant growth and the quantity of yield [3,4,5]. Drought tolerance refers to the adaptions of physiology and biochemistry that enable plants to cope with water deficits [6]. The mechanisms of drought tolerance mainly include assimilated accumulation and translocation, osmotic adjustment, and maintenance of cell wall elasticity [3]. However, these mechanisms can be affected by endophyte infection in grasses [7].

Endophyte associations with cool-season grass species are very common, and the most intensively studied endophytes are asexual members of the genus *Epichloë*, which were previously classified as *Neotyphodium* spp. [8,9]. More than 40 unique taxa in genus *Epichloë*, which form associations with grasses in the subfamily Pooideae, have been described [8]. The associations between endophytes and their plant hosts are generally considered mutualistic [10]. The host plants provide nutrients to the endophytes and, in return, the endophytes can increase the resistance of host plants to biotic and abiotic stresses [7,11,12], such as salt [13], heat [14], heavy metal [15], insects [16], waterlogging [17], and drought [18]. The studies related to the benefits of symbiotic fungal *Epichloë* endophytes for host grasses have mainly focused on the *Lolium* and *Festuca* spp., but there were few studies related to *Festuca sinensis.*

*F. sinensis* is an important native cool-season perennial grass species, which is widely distributed across the cool and semi-arid regions of China [19]. It is becoming an increasingly important species in grassland management and ecological construction in the Qinghai–Tibetan Plateau of China [20,21]. This grass species is frequently symbiotic with an asexual symptomless *Epichloë* endophyte [22,23]. Previous studies have found that *F. sinensis* infected with this endophyte enhanced host fitness by improving seed germination, seedling growth, and tolerance to cold and disease [21,22,23,24,25]. This endophyte has been isolated and identified by morphology with colony, texture, conidia and conidiophore, and phylogeny with the housekeeping gene, which confirmed that the strain is a new species—*Epichloë sinensis* [26]. Recently, Wang et al. [27] have reported that *E. sinensis* could enhance host plant resistance to drought; however, the physiological and biochemical mechanisms related to drought resistance have not been explored. In addition, the resistance of plants to drought may also be related to the genotype that is reflected in the plant ecotype [7]. To comprehensively understand the mechanism by which *E. sinensis* enhances the resistance of *F. sinensis* to drought stress, the present study examines the morphology and physiological variations of two different *F. sinensis* ecotypes, with (E+) and without endophytes (E−), under different soil moisture conditions.

## 2. Results

### 2.1. Plant Height and Tiller Number

Soil moisture significantly (*p* < 0.05) influenced the plant height and tiller number of *F. sinensis*. Compared to 65% RSMC, the plant height and tiller number of both *F. sinensis* ecotypes significantly (*p* < 0.05) decreased at 35% RSMC, with a relative decrease of 33.2% and 44.4%, respectively (Figure 1a,b).

The presence of endophytes significantly (*p* < 0.05) increased the plant height and tiller number of both *F. sinensis* ecotypes under both soil moisture conditions (Figure 1a,b). E+ plants of ecotype 111 and 141 had significantly (*p* < 0.05) higher heights than the corresponding E− plants, with a relative increase of 16.1% and 25.3% at 65% RSMC, and 22.8% and 22.0% at 35% RSMC, respectively (Figure 1a). E+ plants of ecotype 111 and 141 had significantly (*p* < 0.05) higher tiller numbers than the corresponding E− plants, with a relative increase of 21.1% and 28.4% at 65% RSMC, and 86.4% and 35.9% at 35% RSMC, respectively (Figure 1b).

Host ecotypes had significant (*p* < 0.05) impacts on plant height and tiller number under both soil moisture conditions. Ecotype 141 had significantly (*p* < 0.05) higher plant heights than ecotype 111 under 35% RSMC, with a relative increase of 18.1% (Figure 1a). In addition to this, ecotype 141 had significantly (*p* < 0.05) higher tiller numbers than ecotype 111 under both 65% and 35% RSMC, with a relative increase of 21.4% at 65% RSMC and 46.1% at 35% RSMC, respectively (Figure 1b).

### 2.2. Root Indices

Soil moisture significantly (*p* < 0.05) influenced the root indices of *F. sinensis*. Compared to 65% RSMC, the length, volume, and tip number of the roots of both *F. sinensis* ecotypes significantly (*p* < 0.05) decreased at 35% RSMC, with a relative decrease of 32.1%, 36.9%, and 43.9%, respectively (Figure 2a–c).

The presence of endophytes significantly (*p* < 0.05) increased root length and volume of both *F. sinensis* ecotypes under the two soil moisture conditions (Figure 2a,b). E+ plants of ecotypes 111 and 141 had significantly (*p* < 0.05) higher root lengths than the corresponding E− plants, with a relative increase of 15.8% and 19.8% at 65% RSMC, and 47.6% and 24.1% at 35% RSMC, respectively (Figure 2a). E+ plants of ecotypes 111 and 141 had significantly (*p* < 0.05) higher root volumes than the corresponding E− plants (*p* < 0.05), with a relative increase of 24.5% and 24.6% at 65% RSMC, and 37.1% and 16.8% at 35% RSMC, respectively (Figure 2b).

Host ecotypes had significant (*p* < 0.05) impact on root length and root tip number under both soil moisture conditions. Ecotype 141 had significantly (*p* < 0.05) higher root lengths and tip numbers than ecotype 111, with a relative increase of 8.5% and 16.4% at 65% RSMC, and 27.1% and 22.3% at 35% RSMC, respectively (Figure 2a,c).

### 2.3. Biomass of Shoot and Root

Soil moisture significantly (*p* < 0.05) influenced the dry weight of the shoot and root of *F. sinensis*. Compared to 65% RSMC, the dry weights of the shoot and root from both *F. sinensis* ecotypes had significantly decreased at 35% RSMC, with a relative decrease of 36.9% and 34.1%, respectively (Figure 3a,b).

The presence of endophytes significantly (*p* < 0.05) increased the shoot dry weight of E+ plants from both ecotypes 111 and 141, with a relative increase of 12.2% and 14.6% at 65% RSMC, and 21.4% and 16.6% at 35% RSMC, respectively. However, the difference in root dry weights between the E+ and E− plants from ecotypes 111 and 141 was only significant (*p* < 0.05) at 65% RSMC. Under this condition, the E+ plants of ecotype 111 and 141 were higher than the E− plants of the same ecotype, with a relative increase of 14.1% and 15.9%, respectively (Figure 3b). The plants of ecotype 141 had significantly (*p* < 0.05) higher shoot dry weights than ecotype 111, with a relative increase of 14.9% at 65% RSMC and 25.9% at 35% RSMC (Figure 3a).

### 2.4. Chlorophyll Content

Soil moisture significantly (*p* < 0.05) influenced the chlorophyll content of *F. sinensis*. Compared to 65% RSMC, the chlorophyll content of *F. sinensis* significantly (*p* < 0.05) decreased at 35% RSMC, with a relative decrease of 24.5%, respectively (Figure 4).

The presence of endophytes significantly (*p* < 0.05) increased the chlorophyll content of E+ plants from both ecotype 111 and 141, with a relative increase of 8.4% and 8.2% at 65% RSMC, and 9.4% and 7.9% at 35% RSMC, respectively. The plants of ecotype 141 had significantly higher chlorophyll contents than ecotype 111 (*p* < 0.05), with a relative increase of 9.1% at 65% RSMC. Nevertheless, the difference in chlorophyll content between ecotypes 111 and 141 at 35% RSMC was not significant (*p* > 0.05; Figure 4).

Host ecotypes had significant (*p* < 0.05) impact on the chlorophyll content of *F. sinensis* at 65% RSMC. Ecotype 141 had significantly (*p* < 0.05) higher chlorophyll contents than ecotype 111, with a relative increase of 32.5% (Figure 4).

### 2.5. Photosynthetic Indices

Soil moisture significantly (*p* < 0.05) influenced the photosynthetic indices of *F. sinensis.* Compared to 65% RSMC, three photosynthetic indices, including the net photosynthetic rate, stomatal conductance, and transpiration rate of E+ and E− plants, of both ecotype 111 and 141, were significantly (*p* < 0.05) decreased at 35% RSMC, with a relative decrease of 32.0%, 44.9%, and 38.6%, respectively, whereas the intercellular carbon dioxide concentration had the opposite trend and showed a relative increase of 56.4% (Figure 5a–d).

The presence of endophytes significantly (*p* < 0.05) increased the net photosynthetic rate of E+ plants from ecotype 111 and 141, with a relative increase of 16.4% and 33.9% at 65% RSMC, and 14.8% and 8.7% at 35% RSMC, respectively. The plants of ecotype 141 had significantly (*p* < 0.05) higher net photosynthetic rates than ecotype 111, with a relative increase of 13.3% at 65% RSMC and 8.7% at 35% RSMC (Figure 5a). The stomatal conductance of E+ plants from ecotype 111 and 141 were significantly (*p* < 0.05) higher than E− plants of the same ecotype, with a relative increase of 17.6% and 40.6% at 65% RSMC, and 12.7% and 34.1% RSMC at 35% RSMC, respectively. The plants of ecotype 141 had a significantly (*p* < 0.05) higher stomatal conductance than ecotype 111, with a relative increase of 12.5% at 65% RSMC and 8.3% at 35% RSMC (Figure 5b). Similarly, the presence of endophytes significantly (*p* < 0.05) increased the transpiration rate of E+ plants from ecotype 111 and 141, with a relative increase of 34.8% and 33.3% at 65% RSMC, and 12.1% and 22.7% at 35% RSMC, respectively. The plants of ecotype 141 had significantly (*p* < 0.05) higher transpiration rates than ecotype 111, with a relative increase of 16.7% at 65% RSMC and 15.7% at 35% RSMC (Figure 5c). Conversely, the presence of endophytes significantly decreased (*p* < 0.05) the intercellular carbon dioxide concentration of E+ plants of ecotype 111 and 141, with a relative decrease of 19.5% and 38.5% at 65% RSMC, and 16.2% and 12.1% at 35% RSMC, respectively. The plants of ecotype 111 had significantly (*p* < 0.05) lower intercellular carbon dioxide concentrations than ecotype 141, with a relative decrease of 6.7% at 65% RSMC and 5.9% at 35% RSMC (Figure 5d).

### 2.6. Phytohormones

The contents of phytohormones, including ABA, CTK, and GA, were significantly (*p* < 0.05) influenced by soil moisture conditions. Compared to 65% RSMC, the ABA content had significantly (*p* < 0.05) increased at 35% RSMC, with a relative increase of 25.6%; whereas, the CTK and GA contents had significantly (*p* < 0.05) decreased at 35% RSMC, with a relative decrease of 19.9% and 21.6%, respectively (Figure 6a,c,d).

The presence of endophytes significantly (*p* < 0.05) increased the ABA content of E+ plants from ecotypes 111 and 141, with a relative increase of 6.2% and 14.4% at 35% RSMC. The plants of ecotype 111 had significantly (*p* < 0.05) higher ABA contents than ecotype 141, with a relative increase of 8.0% at 65% RSMC and 30.3% at 35% RSMC, respectively (Figure 6a). On the contrary, the presence of endophytes significantly (*p* < 0.05) decreased the CTK content of E+ plants from ecotype 111 and 141, with a relative decrease of 20.1% and 18.2% at 35% RSMC; meanwhile, at 65% RSMC, the endophyte only significantly (*p* < 0.05) decreased the CTK content of ecotype 111, with a relative decrease of 11.7%. The plants of ecotype 141 had significantly (*p* < 0.05) higher CTK contents than ecotype 111, with a relative increase of 13.1% at 65% RSMC and 9.8% at 35% RSMC, respectively (Figure 6c).

### 2.7. Contents of K^+^ and Ca^2+^

Soil moisture significantly (*p* < 0.05) influenced the Ca^2+^ content of *F. sinensis*. Compared to 65% RSMC, the Ca^2+^ content of *F. sinensis* significantly (*p* < 0.05) decreased at 35% RSMC, with a relative decrease of 20.4% (Figure 7b).

The presence of endophytes significantly (*p* < 0.05) increased the K^+^ content of E+ plants from ecotype 111 and 141 at 35% RSMC, with a relative increase of 19.8% and 11.7%, respectively. At 65% RSMC, the endophytes only significantly (*p* < 0.05) increased the K^+^ content of E+ plants from ecotype 111, with a relative increase of 8.1% (Figure 7a). Similarly, the presence of endophytes significantly increased the Ca^2+^ content of E+ plants from ecotypes 111 and 141 at 35% RSMC (*p* < 0.05), with a relative increase of 19.8% and 45.5%, respectively. At 65% RSMC, the endophytes only significantly increased the Ca ^2+^ content of E+ plants from ecotype 111 (*p* < 0.05), with a relative increase of 8.3% (Figure 7b).

Host ecotypes had significant (*p* < 0.05) impacts on the Ca^2+^ content of *F. sinensis.* The plants of ecotype 141 had significantly higher Ca^2+^ contents than ecotype 111 (*p* < 0.05), with a relative increase of 11.4% at 65% RSMC and 26.9% at 35% RSMC (Figure 7b).

### 2.8. Contents of C, N, and P

The contents of C, N, and P were significantly influenced by soil moisture conditions, and these were significantly higher at 65% RSMC than that at 35% RSMC (*p* < 0.05) in both the shoots and roots (Figure 8a–f). The presence of endophytes significantly increased the C content of E+ plants from ecotypes 111 and 141 in the shoots (*p* < 0.05), with a relative increase of 10.2% and 12.9% at 65% RSMC, and 13.1% and 14.9% at 35% RSMC, respectively. However, the C content of E+ plants from ecotypes 111 and 141 in the roots only was significantly higher than that of the E− plants at 65% RSMC, with a relative increase of 12.7% and 8.2%, respectively (Figure 8a,b). Similarly, the presence of endophytes significantly increased the N content of E+ plants from ecotypes 111 and 141 in the shoots (*p* < 0.05), with a relative increase of 8.6% and 15.3% at 65% RSMC, and 16.1% and 10.6% at 35% RSMC, respectively. On the other hand, the N content of E+ plants from ecotypes 111 and 141 in the roots was the only content that was significantly higher than E− plants at 65% RSMC, with a relative increase of 17.4% and 15.6%, respectively (Figure 8c,d). However, the presence of endophytes significantly increased the P content of E+ plants from both ecotype 111 and 141 (*p* < 0.05), regardless of them being in 65% or 35% RSMC and regardless of them being in the shoots or roots (Figure 8e,f). In the shoots, the P content of E+ plants of both ecotype 111 and 141 was significantly higher than E− plants (*p* < 0.05), with a relative increase of 27% and 13.9% at 65% RSMC, and 19.1% and 6.5% at 35% RSMC. Additionally, in the shoots, the P content of E+ plants of both ecotype 111 and 141 was also significantly higher than E− plants (*p* < 0.05), with a relative increase of 27.3% and 11.1% at 65% RSMC and 22.5% and 18.3% at 35% RSMC. In the C, N, and P contents between the two ecotypes, there was only a significant difference in the C content in the shoot at 65% RSMC; the ecotype 141 had a significantly higher C content than ecotype 111 (*p* < 0.05), with a relative increase of 10.6% (Figure 8a).

### 2.9. Ecological Stoichiometry

The C:P ratio and N:P ratio were significantly influenced by soil moisture conditions, and these ratios were significantly higher at 35% RSMC than that at 65% RSMC (*p* < 0.05) in the shoot (Figure 9c,e). By contrast, the C:P ratio and N:P ratio in the root and C:N ratio in both the shoot and root was not influenced by soil moisture conditions (Figure 9a,b,d,f).

The presence of endophytes significantly increased the C:N ratio of ecotype 141 in both the shoot and the root at 65% RSMC, as well as in the shoot of ecotype 111 at 35% RSMC (*p* < 0.05; Figure 9a,b). The presence of endophytes significantly increased the C:P ratio of ecotype 111 in both shoot and root at 65% RSMC and in root at 35% RSMC, while the C:P ratio of ecotype 141 was only significantly influenced by endophytes in the root at 35% RSMC (*p* < 0.05) (Figure 9c,d). The presence of endophytes also significantly increased the N:P ratio of ecotype 111 in both the shoot and the root at 65% RSMC, and in the root at 35% RSMC, as well as in the root of ecotype 141 at 35% RSMC (*p* < 0.05; Figure 9e,f).

The host ecotype had significant (*p* < 0.05) impacts on the C:N ratio, C:P ratio, and N:P ratio. The ecotype 141 had a significantly higher C:N ratio than ecotype 111 at both 65% and 35% RSMC in the shoot, and at 35% RSMC in the root (*p* < 0.05) (Figure 9a,b). However, the C:P ratio was only significantly different between ecotypes 111 and 141 in the shoot at 65% RSMC (Figure 9c,d). Similarly, with the N:P ratio, the ecotype 141 had a significantly higher N:P ratio than ecotype 111 in both the shoot and the root at 65% RSMC (*p* < 0.05), but ecotype 111 had a significantly higher N:P ratio than ecotype 141 in both the shoot and the root at 35% RSMC (*p* < 0.05; Figure 9e,f).

## 3. Discussion

In this study, we combined various indices relating to drought tolerance from two *F. sinensis* ecotypes to comprehensively evaluate the difference between E+ and E− plants under different soil moisture conditions (35% RSMC and 65% RSMC). The results showed that the growth and photosynthesis of *F. sinensis* were inhibited at 35% RSMC, suggesting that the water deficit had a great influence on plant physiology. The previous studies about the response of *F. sinensis* to different water levels also confirmed that 35% RSMC was considered drought stress for these two ecotypes of *F. sinensis* [27].

Many studies have demonstrated that *Epichloë* endophytes can enhance host plant resistance to drought stress by improving growth and biomass [28,29,30], controlling stomatal status [31,32,33] and adjusting the cell osmotic potential [11,34,35]. However, while these studies mainly focused on the endophytes of *Lolium* and *Festuca* spp., little is known of *F. sinensis*. Recently, the endophyte associated with *F. sinensis* was identified as a new *Epichloë* species—*E. sinensis* [26]. In our case, it is necessary to conduct more studies about the effects of *E. sinensis* on the drought resistance of its host, *F. sinensis.*

The root system, being the main water-absorbing organ, is very important for plant growth, especially in arid environments [36]. The endophytes improved drought resistance mainly by promoting the growth and development of host roots, even though the endophyte was not found in the root of host plants [29,37]. In our study, the endophyte infection significantly increased the growth indices of host plants under drought stress (including the growth of aboveground and underground parts). This is similar to the findings of Nagabhyru et al. [38] and Hesse et al. [39]. Our results are also consistent with our previous studies on *F. sinensis* [27], confirming that endophytes can enhance the drought resistance of its host, *F. sinensis*, by improving the growth of the host plant.

Photosynthesis in plants is directly related to the accumulation of biomass, and it is also one of the most important physiological processes in plants that are most vulnerable to water deficit [40]. Liang et al. [41] found that the net photosynthetic rate of E+ plants under severe drought was significantly higher than that of E− plants in *Lolium perenne* L. Further, Morse et al. [42] showed that the net photosynthetic rate, stomatal conductance, and transpiration rate of E+ plants were significantly higher than those of the E− plants under severe drought stress in *Festuca*. In our study, endophyte infection not only increased chlorophyll concentration but also improved all of the photosynthetic indices of *F. sinensis* under drought stress, with the exception of the intercellular carbon dioxide concentration. These results indicate that endophytes can enhance the drought resistance of *F. sinensis* by improving photosynthesis, which was consistent with the result that endophytes can change the strategy of the host plant and enhance the metabolism and accumulation of substances in host plants to adapt to water-deficient conditions [43].

Phytohormones are compounds derived from plant biosynthetic pathways that can act either at the site of synthesis or in following the transportation route [44]. There is increasing evidence showing that the plant physiological response to biotic and abiotic stresses is largely governed by phytohormone balance [45,46]. ABA, the most studied phytohormone, yields one of the fastest responses to abiotic stress in plants; this is especially so for water stress [47]. The accumulation and synthesis of ABA promote stomatal closure to minimize water loss via transpiration [48]. On the contrary, CTK triggers responses to delay stomatal closure [49,50,51] or stimulate stomatal opening [52]. Therefore, drought-resistant plants reduce their CTK levels to increase their chances of survival under drought stress [53,54]. Possibly, CTK and ABA have a relationship that is antagonistic to stress [55,56,57]. In our study, the presence of endophytes increased the ABA content but decreased the CTK content under drought stress. Similarly, researchers have found that endophytes can improve the resistance of *Stipa purpurea* to parasitic stress (*Pedicularis kansuensis*) by increasing the ABA content and decreasing the CTK content [58]. Another research on ryegrass also found that endophytes increased the content of ABA to improve host tolerance to drought stress [59]. These studies indicate that endophytes affect plant resistance to biotic and abiotic stresses by regulating ABA and CTK. In addition, CTK does not always act as an ABA antagonist. The decrease in CTK content may enhance the sensitivity and response to water deficit or increase cell membrane integrity and stability rather than to ABA-mediated stomatal closure [53]. Regulations of plant response to their external environment are accomplished by the synergistic or antagonistic action of various phytohormones [52]. In our study, the IAA and GA contents were also determined. These two phytohormones usually affect plant root development, especially in drought conditions [60,61]. The contents of these two phytohormones were not influenced by endophytes, indicating that endophytes did not regulate these two phytohormones to affect the drought resistance of *F. sinensis.*

Under drought stress, plants can regulate ion absorption and transportation to accumulate inorganic ions for osmotic adjustment and water balance [62]. Research has demonstrated that osmotic adjustment in leaves is closely related to the drought resistance of plants [63]. K^+^ and Ca^2+^ are two important ions related to osmotic adjustment in plants. K^+^, as the most important osmotic regulator, plays a decisive role in maintaining plant cell turgor [64]. When plants suffer from water deficit, K^+^ is often immediately transported to the most vigorous tissues [65]. Bayat et al. [66] found that K^+^ concentration in E+ plants was higher than in E− plants in tall fescue under drought stress. Similarly, Sabzalian and Mirlohi [37] also reported that endophyte infection increased the content of K^+^ in *F. arundinacea* and *F. pratensis* under drought stress. In our study, endophytes significantly increased the K^+^ concentration in *F. sinensis* under drought stress; however, K^+^ concentration was not influenced by soil water conditions. Therefore, we speculate that drought resistance in *F. sinensis* was related to the accumulation of K^+^, which is affected by the presence of endophytes. Ca^2+^ plays an essential role in ion transport regulation, cell wall enzyme activities, cell membrane integrity, cell wall structure stabilization, plant cell metabolism regulation, and plant cell signal transduction [67,68]. Compared to K^+^, Ca^2+^ is not as mobile in the plant, and its movement mainly depends on the mass flow of water [66]. Under drought stress, the affinity of plant roots to calcium decreased, which resulted in a decrease in calcium absorption by the plants. Increasing the concentration of Ca^2+^ can regulate enzymatic activity, including that of SOD (superoxide dismutase), POD (peroxidase), CAT (catalase), and other enzymes, thereby eliminating the oxidative stress damage in plants under drought conditions. In our study, drought stress was found to limit the absorption of Ca^2+^, but the presence of endophytes significantly increased the Ca^2+^ concentration. These results suggest that endophytes promote the resistance of *F. sinensis* to drought by accumulating Ca^2+^ and eliminating the oxidative stress damage caused by drought stress.

Photosynthesis is the main way for green plants to assimilate C, while N and P assimilation occurs through nutrient uptake in soil [69]. In our study, all the contents of C, N, and P in the aboveground and underground parts of the plants significantly decreased under drought conditions, and the results were probably related to photosynthesis. Some studies found that under drought stress, the inhibition of photosynthesis leads to a decline in the ability of plants to assimilate carbon [70] and a decrease in the rate of plant transpiration, which causes the reduced ability of plants to absorb and transport nutrients through the root system [13,71]. This occurrence then leads to a decrease in nitrogen and phosphorus. The results of these studies are consistent with our findings which showed that drought stress inhibited the C, N, and P assimilation of *F. sinensis.*

Endophytes can improve the photosynthesis of host plants under drought conditions (as described above). In this study, the C content of plant aboveground parts increased. Moreover, under drought stress, endophytes were found to improve root growth, but the biomass of the plant’s underground parts was not influenced by endophytes. Combining the fact that the C content in the underground parts of the plant is not affected by endophytes, we speculate that plants transfer a portion of the carbon to its aboveground parts under drought conditions and endophytes play an important role in this process. The element N is not only an essential component of enzymes [72], but it is also an important component of alkaloids, and its concentration is positively correlated with the N content [73,74]. In our study, the presence of endophytes increased the N content in the shoot, and this is most likely related to alkaloid synthesis. Plants produce more alkaloids under drought conditions (as described below); therefore, N elements in the underground parts of the plant may also be transferred to aboveground parts to participate in alkaloid synthesis. N availability potentially promotes P uptake through the stimulation of phosphatase activity in the roots [43,75]. In our study, endophytes increased the P content in both the aboveground and underground parts of *F. sinensis*, indicating that endophytes may improve the growth of host plants under drought stress by absorbing more of the P element.

The C:N:P stoichiometry has been widely applied because it can represent the ability of a plant species to assimilate C when absorbing N and P synchronously and explain phenomena in diverse ecological processes. It has been reported that the ratio of C:N and C:P could reflect the health and growth status of plants and have a negative correlation with the growth rate of plants [76,77,78,79], whereas the N:P ratio might reflect the nutrient limitations during the growth process of plants, especially the dynamic balance between soil nutrients and plant nutrient demands [79,80]. In our study, the presence of endophytes reduced the C:N ratio of aboveground and underground parts in ecotype 141 under optimum water conditions and the C:N ratio of aboveground parts in ecotype 111 under drought stress conditions. At the same time, the presence of endophytes reduced the C:P ratio of aboveground and underground parts of ecotype 111 under optimum water conditions. The C:P ratio of underground parts of ecotype 141 under drought stress conditions was also reduced. According to the growth rate hypothesis (GRH), a higher plant growth rate would be mainly accompanied by a lower ratio of C:N and C:P [13]. The results showed that endophytes could improve the growth of host plants, but their effects were affected by soil moisture and plant ecotypes. In addition, endophytes significantly reduced the N:P ratio in the underground parts of the two ecotypes, which may be due to the transfer of nitrogen elements from underground parts to aboveground parts to participate in alkaloid synthesis and other processes. However, the N:P ratio in the aboveground part of the plant was not influenced by endophytes, and only the N:P ratio of ecotype 111 significantly decreased. These results indicate that both the presence of endophytes and the presence of soil moisture have significant effects on the stoichiometry of host plants. The results also suggest that the adjustment strategies of different host ecotypes are not consistent.

The synthesis and accumulation of small molecular organic solutes, such as proline and soluble sugar, are the main tools of plant osmotic regulation [81,82,83,84]. Studies have shown that when plants are subjected to drought stress, proline in the symbionts can be converted into precursor compounds of loline or ergot alkaloids [85,86,87]. Therefore, changes in alkaloid content can also be considered as a means of osmotic regulation [88,89,90]. Several previous studies reported that the concentration of alkaloids increased in response to increasing water stress [11,89,91,92]. So far, three types of alkaloids, including peramine, lolitrem B, and ergot, have been tested in endophyte—*F. sinensis* symbionts [21,25]. Therefore, the relationship between the alkaloids content change in *F. sinensis* and the drought tolerance of *F. sinensis* under water stress need to be further studied.

The effects of endophytes on the drought resistance of host plants were mainly dependent on environmental conditions and host plant genotype [93,94,95]. Hesse et al. [39,96] found that water deficiency triggered positive endophytic effects on the seed production of *L. perenne*, but these effects were not observed in the genotype originating from a wet site, and the genotype that originated from a wet site showed higher sensitivity to drought stress when endophyte infection was present. In our study, endophytes improved the drought resistance of both *F. sinensis* ecotypes, but there was a significant difference between the two ecotypes. Compared to ecotype 111, all of the growth and photosynthetic indices of ecotype 141 were better under both treatments. This is consistent with the fact that the shoot dry weight of ecotype 141 was higher than that of ecotype 111 under both treatments. Although a higher Ca^2+^ concentration in ecotype 141 indicated a stronger ability to repair damage caused by drought stress, a higher ABA content and lower CTK content in ecotype 111 indicated a stronger sensitivity level and a faster response to water deficit [52]. The above results provide a reference for the breeding of drought-resistant varieties *F. sinensis,* and, in terms of growth performance alone, ecotype 141 was found to be better than ecotype 111.

In conclusion, the results of this study demonstrate that *E. sinensis* enhances the drought resistance of host *F. sinensis* by improving the growth and photosynthesis, accumulating K^+^ and Ca^2+^, and regulating phytohormones of ABA and CTK. Our study reveals the mechanism that endophytes use to improve the drought tolerance of *F. sinensis* and provide a foundation for breeding programs using endophytes.

## 4. Materials and Methods

### 4.1. Plant Materials

The seeds of two *F. sinensis* ecotypes were originally collected from the natural grasslands of Hongyuan County in Sichuan Province, China (32°48′ N, 102°33′ E, Altitude 3491 m, ecotype 111) and Ping’an County in Qinghai Province, China (36°19′ N, 101°51′ E, Altitude 2922 m, ecotype 141) in the summer of 2013. The E+ and E− population of these two ecotypes were set up in the experimental field of the Yuzhong Pastoral Agriculture Experimental Station of Lanzhou University (35°56′ N, 104°08′ E, Altitude 1718 m) as described in Tian et al. [26]. The seeds of both ecotypes of *F. sinensis* were harvested from E+ and E− plants in 2015 and stored at a constant temperature of 4 °C in darkness before utilization.

### 4.2. Experimental Design

On April 21, 2017, seeds of these two ecotypes were, respectively, planted into 72-cell plastic seedling trays containing sterilized vermiculite. These were then grown under constant greenhouse conditions (temperature: 22 °C ± 2 °C, moisture: 46% ± 2%) at the College of Pastoral Agriculture Science and Technology, Lanzhou University (35°36′ N, 105°26′ E, Altitude 1718 m). The endophyte infection statuses of these seedlings were determined by aniline blue staining [97] after the seedlings developed 3–5 tillers. The seedlings germinated from E+ seeds with endophyte hyphae were marked as E+ and the seedlings germinated from E− seeds without endophyte hyphae were marked as E−. Thereafter, all seedlings were transplanted from seedling trays to pots containing the same amount of sterilized peat soil and vermiculite (weight ration 3:1). Each pot (diameter: 24 cm; height: 14 cm) had only one seedling and underwent the same irrigation regime to maintain a uniform soil moisture content.

A pot experiment was performed from 20 June to 5 August in the greenhouse. On 20 June, 10 E+ plants and 10 E− plants per ecotype that were of similar sizes were selected. Each pot was weighed and irrigated separately every day at 6 p.m. until the end of this experiment to maintain the soil moisture content at 35% or 65% RSMC. The 35% RSMC was the drought stress treatment, and 65% RSMC was the control treatment (based on the research of Wang et al. [27]). Each treatment had five replicates, and the position of each pot was changed randomly and regularly until the end of this experiment.

### 4.3. Measurement Protocols

#### 4.3.1. Plant Performance

On August 5, all plants were destructively harvested and carefully removed from the pots. The whole plants were cleaned with distilled water and wiped off with filter paper before the plant height, tiller number, and root indices of each plant were determined. The root volume was measured using the displacement technique with a graduated cylinder [98]. The root tip number was counted manually. Thereafter, all the plants were separated into shoot and root components and oven-dried at 80 °C until a constant weight was reached. After weighing, the plant materials were ground twice using a mixer mill (MM 400; Retsch, Haan, Germany) at 30 Hz for 2 min for the analysis of ion contents, nutrient elements, and alkaloids.

#### 4.3.2. Chlorophyll Content

On August 4 (six weeks after the water treatment), a chlorophyll meter (SPAD-502Plus; Konica Minolta Sensing Inc., Qsaka, Japan), was used to measure the chlorophyll content within the top leaf of each plant. Four measurements were performed at four points along the selected leaf of each plant.

#### 4.3.3. Photosynthetic Indices

On August 4, the photosynthetic indices, including net photosynthesis rate, stomatal conductance, transpiration rate, and intercellular carbon dioxide concentration, were measured using the portable photosynthesis system (Li-6400; LI-COR Inc., Lincoln, NE, USA) from 9:00 a.m. to 11:00 a.m. Photosynthesis in the top leaf of each plant in each treatment was measured. Four measurements were performed at four points along each leaf on each plant.

#### 4.3.4. Phytohormones

On August 4, leaves were cut at the same site of the plants as samples to determine the concentrations of phytohormones, including indolE−3-acetic (IAA), abscisic acid (ABA), cytokinin (CTK), and gibberellic acid (GA). The samples were wrapped with aluminum foil and temporarily stored in liquid nitrogen before being stored at −80 °C. The plant hormones were determined by a plant enzymE−linked immunosorbent assay kit (GR-E91649, E91056, E91106, E91022; Shanghai Huyu Biotechnology Co. Ltd., Shanghai, China).

#### 4.3.5. Potassium (K^+^) and Calcium (Ca^2+^) Ion Contents

Plant ion contents were determined following digestion with H_2_SO_4_ at a temperature of 400 °C. The atomic absorption spectrometer (FS95 Furnace Autosampler; Graphite Bio, Inc., South San Francisco, CA, USA) was used to determine the contents of potassium and calcium ions [99].

#### 4.3.6. C, N, and P Contents

Total organic C content was determined with the oil bath K_2_CrO_7_ titration method (oxidization with dichromate in the presence of H_2_SO_4_, heated at 180 °C for 5 min, and titrated with FeSO_4_). The total N and P contents in the samples were determined following digestion with H_2_SO_4_ at a temperature of 420 °C. The N and P concentrations in the digested solutions were determined by flow injection analysis, using an Analyzer (FIAStar 5000, Foss Tecator, Hoganas, Sweden) [43].

### 4.4. Statistical Analysis

Statistical data analysis was performed with the SPSS Inc. software (Released 2009. PASW Statistics for Windows, Version 18.0., Chicago, IL, USA). The differences in all indices in this experiment were analyzed by the independent *t*-test and repeated-measures ANOVA with Fisher’s least significant differences (LSD) test. These included the differences between E+ and E− plants of the same ecotype, or ecotype 111 and 141 under the same soil moisture conditions, or all plants between the two treatments (65% and 35% RSMC). Statistical significance was defined at the 95% confidence level, and means were reported with their standard error.

## Figures and Tables

**Figure 1 plants-10-01649-f001:**
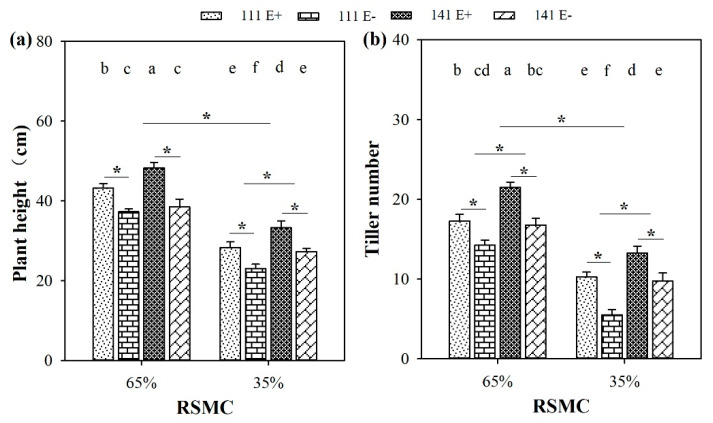
Plant height (**a**) and tiller number (**b**) of E+ and E− *F. sinensis* of ecotype 111 and 141 under two different soil moisture conditions (65% and 35%). Values are the mean ± standard error. * Represents significant difference at *p* < 0.05 level (independent *t*-test) between E+ and E− plants of the same ecotype under the same soil moisture conditions, or between ecotype 111 and 141 under the same soil moisture conditions or all plants between two treatments 65% and 35% RSMC (If appeared), different lowercase letters above boxes represents significant difference at *p* < 0.05 level between E+ and E− plants of two ecotypes under both 65% and 35% RSMC.

**Figure 2 plants-10-01649-f002:**
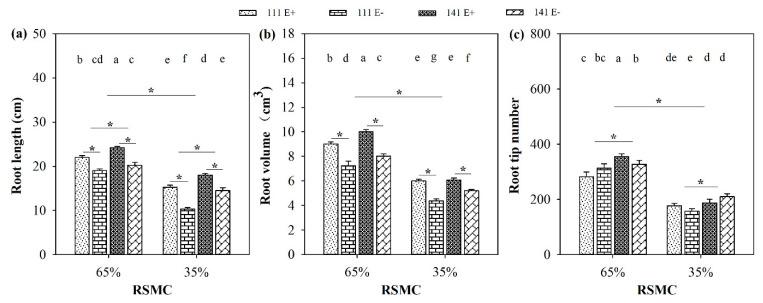
Root length (**a**), volume (**b**) and tip number (**c**) of E+ and E− *F. sinensis* from ecotype 111 and 141 under two different soil moisture conditions. Values are the mean ± standard error. * Represents significant difference at *p* < 0.05 level (independent *t*-test) between E+ and E− plants of the same ecotype under the same soil moisture conditions, or between ecotype 111 and 141 under the same soil moisture conditions or all plants between two treatments 65% and 35% RSMC (If appeared), different lowercase letters above boxes represents significant difference at *p* < 0.05 level between E+ and E− plants of two ecotypes under both 65% and 35% RSMC.

**Figure 3 plants-10-01649-f003:**
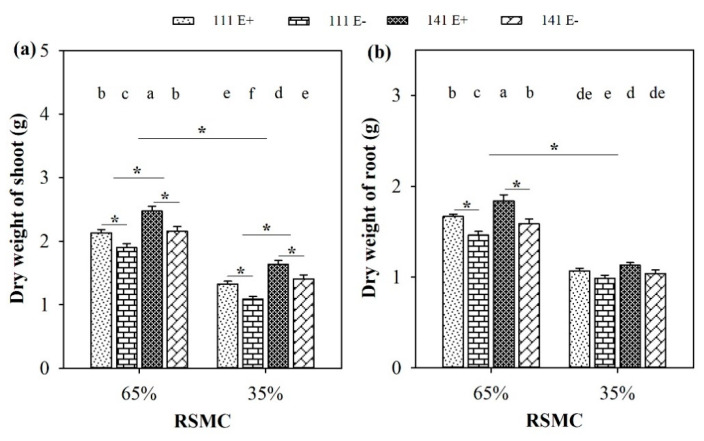
Dry weight of shoot (**a**) and root (**b**) of E+ and E− *F. sinensis* from ecotype 111 and 141 under two different soil moisture conditions. Values are the mean ± standard error. * Represents significant difference at *p* < 0.05 level (independent *t*-test) between E+ and E− plants of the same ecotype under the same soil moisture conditions, or between ecotype 111 and 141 under the same soil moisture conditions or all plants between two treatments 65% and 35% RSMC (If appeared), different lowercase letters above boxes represents significant difference at *p* < 0.05 level between E+ and E− plants of two ecotypes under both 65% and 35% RSMC.

**Figure 4 plants-10-01649-f004:**
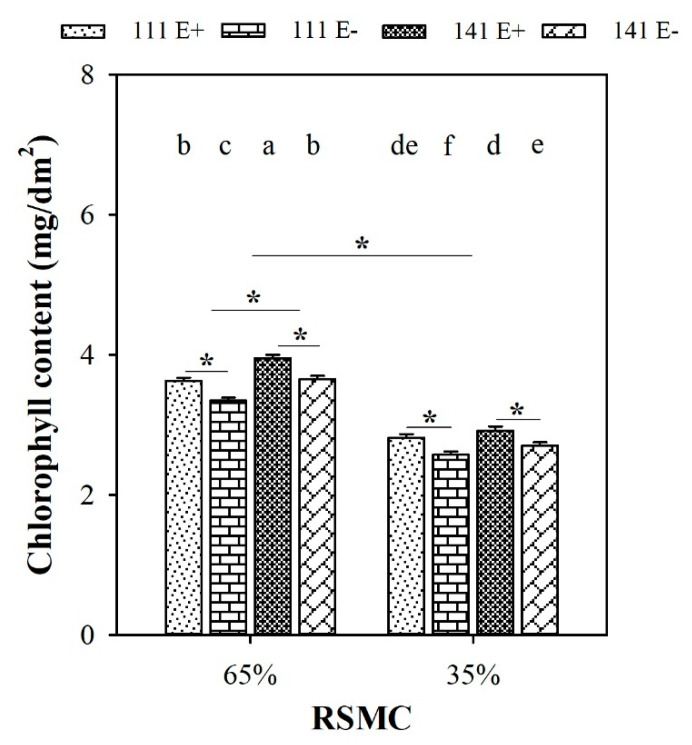
The chlorophyll content of E+ and E− *F. sinensis* from ecotype 111 and 141 under two different soil moisture conditions. Values are the mean ± standard error. * Represents significant difference at *p* < 0.05 level (independent *t*-test) between E+ and E− plants of the same ecotype under the same soil moisture conditions, or between ecotype 111 and 141 under the same soil moisture conditions or all plants between two treatments 65% and 35% RSMC (If appeared), different lowercase letters above boxes represents significant difference at *p* < 0.05 level between E+ and E− plants of two ecotypes under both 65% and 35% RSMC.

**Figure 5 plants-10-01649-f005:**
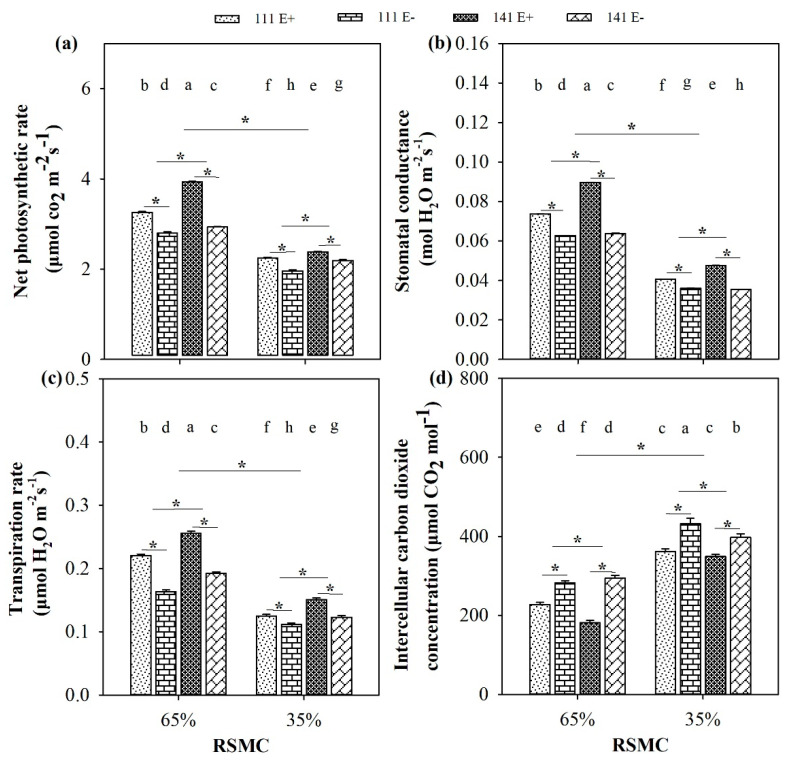
The photosynthetic indices (**a**–**d**) of E+ and E− *F. sinensis* from ecotype 111 and 141 under two different soil moisture conditions. Values are the mean ± standard error. * Represents significant difference at *p* < 0.05 level (independent *t*-test) between E+ and E− plants of the same ecotype under the same soil moisture conditions, or between ecotype 111 and 141 under the same soil moisture conditions or all plants between two treatments 65% and 35% RSMC (If appeared), different lowercase letters above boxes represents significant difference at *p* < 0.05 level between E+ and E− plants of two ecotypes under both 65% and 35% RSMC.

**Figure 6 plants-10-01649-f006:**
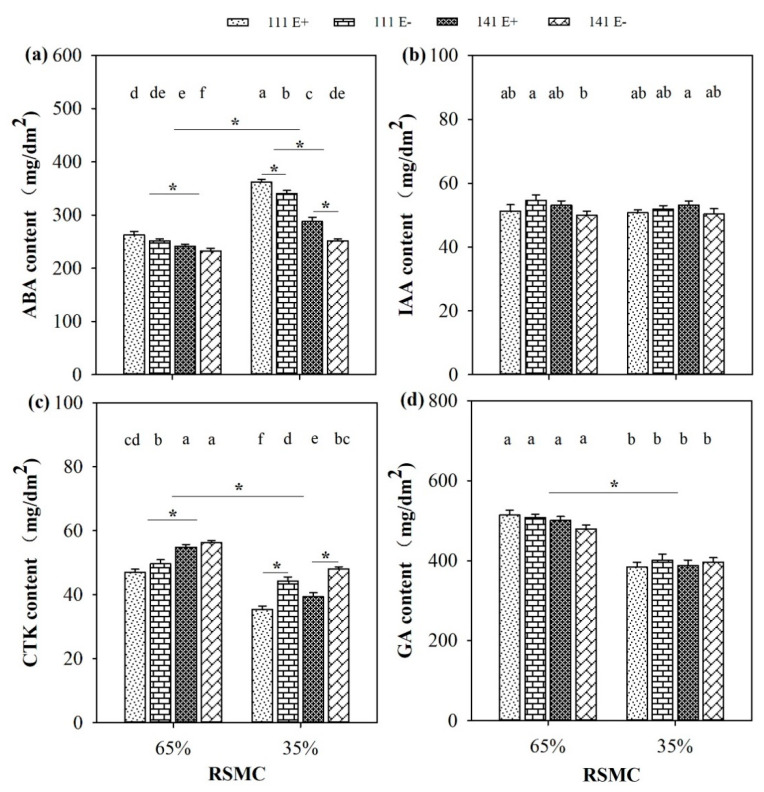
The phytohormones (**a**–**d**) of E+ and E− *F. sinensis* from ecotype 111 and 141 under two different soil moisture conditions. Values are the mean ± standard error. * Represents significant difference at *p* < 0.05 level (independent *t*-test) between E+ and E− plants of the same ecotype under the same soil moisture conditions, or between ecotype 111 and 141 under the same soil moisture conditions or all plants between two treatments 65% and 35% RSMC (If appeared), different lowercase letters above boxes represents significant difference at *p* < 0.05 level between E+ and E− plants of two ecotypes under both 65% and 35% RSMC.

**Figure 7 plants-10-01649-f007:**
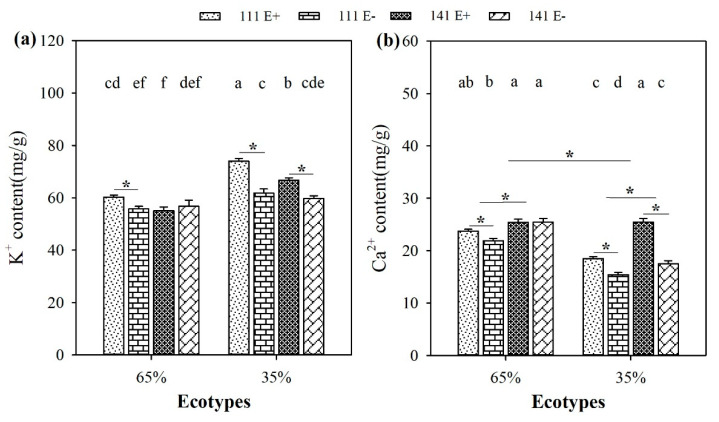
The content of K^+^ (**a**) and Ca^2+^ (**b**) of E+ and E− *F. sinensis* from ecotype 111 and 141 under two different soil moisture conditions. Values are the mean ± standard error. * Represents significant difference at *p* < 0.05 level (independent *t*-test) between E+ and E− plants of the same ecotype under the same soil moisture conditions, or between ecotype 111 and 141 under the same soil moisture conditions or all plants between two treatments 65% and 35% RSMC (If appeared), different lowercase letters above boxes represents significant difference at *p* < 0.05 level between E+ and E− plants of two ecotypes under both 65% and 35% RSMC.

**Figure 8 plants-10-01649-f008:**
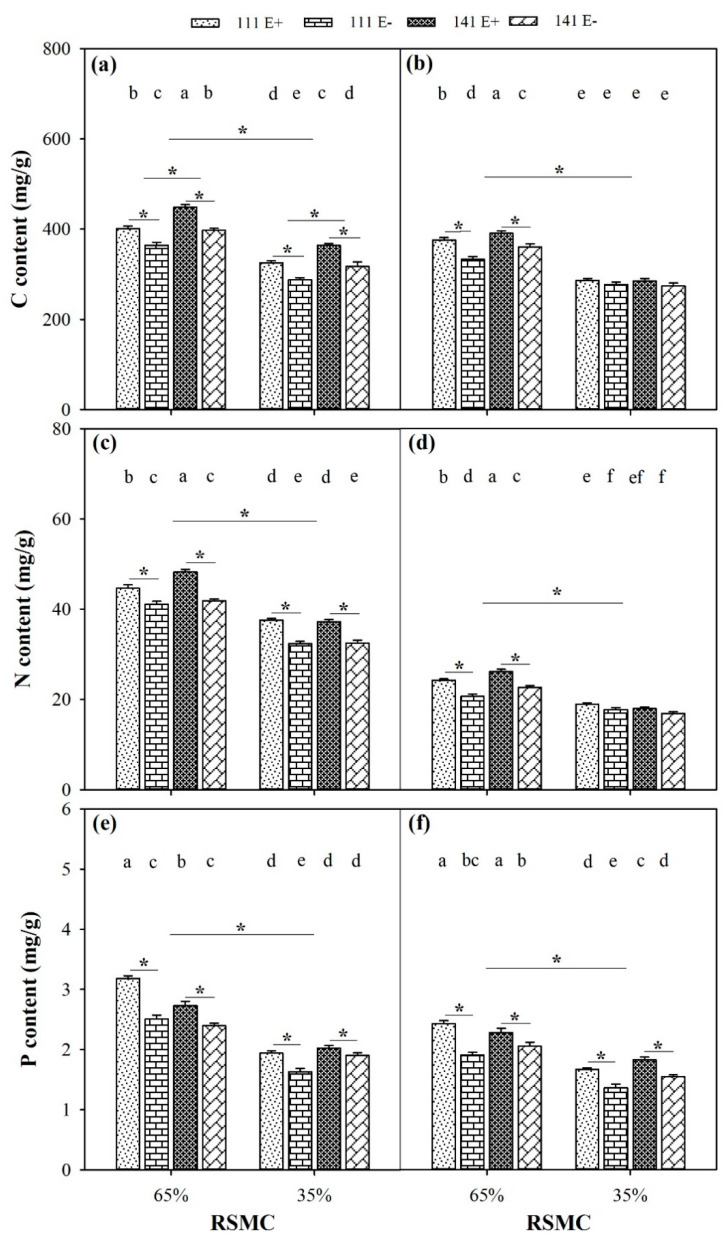
The content of C, N and P in shoots (**a**,**c**,**e**) and roots (**b**,**d**,**f**) of E+ and E− *F. sinensis* from ecotype 111 and 141 under two different soil moisture conditions. Values are the mean ± standard error. * Represents significant difference at *p* < 0.05 level (independent *t*-test) between E+ and E− plants of the same ecotype under the same soil moisture conditions, or between ecotype 111 and 141 under the same soil moisture conditions or all plants between two treatments 65% and 35% RSMC (If appeared), different lowercase letters above boxes represents significant difference at *p* < 0.05 level between E+ and E− plants of two ecotypes under both 65% and 35% RSMC.

**Figure 9 plants-10-01649-f009:**
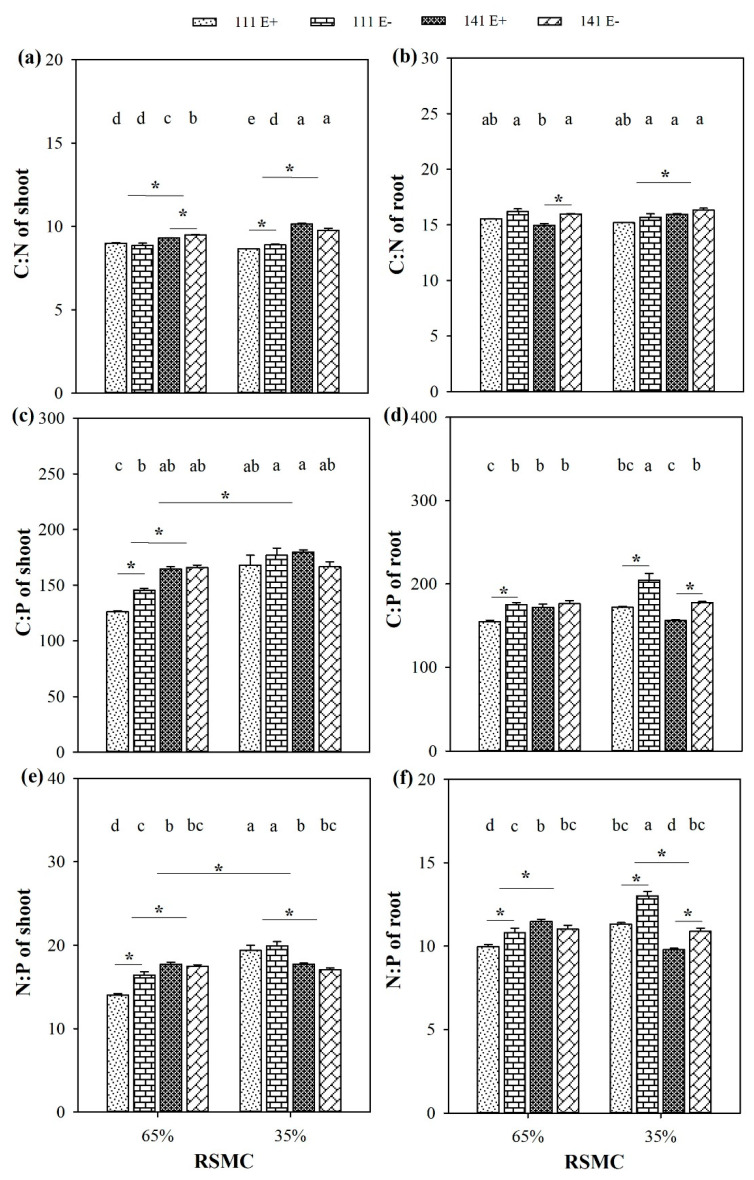
The ratio of C:N, C:P and N:P in shoots (**a**,**c**,**e**) and roots (**b**,**d**,**f**) of E+ and E− *F. sinensis* from ecotype 111 and 141 under two different soil moisture conditions. Values are the mean ± standard error. * Represents significant difference at *p* < 0.05 level (independent *t*-test) between E+ and E− plants of the same ecotype under the same soil moisture conditions, or between ecotype 111 and 141 under the same soil moisture conditions or all plants between two treatments 65% and 35% RSMC (If appeared), different lowercase letters above boxes represents significant difference at *p* < 0.05 level between E+ and E− plants of two ecotypes under both 65% and 35% RSMC.

## Data Availability

Not applicable.

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
