# Peer review of "Effects of Epichloë sinensis Endophyte and Host Ecotype on Physiology of Festuca sinensis under Different Soil Moisture Conditions"

_plants, 2021, doi:10.3390/plants10081649_

Round 1

Reviewer 1 Report

The current article titled Effects of Epichloë sinensis endophyte and host ecotype on physiology of Festuca sinensis under different soil moisture conditions, submitted for publication by Wenbo Xu et al., describes the effects of the endophyte, Epichloë sinensis on growth, photosynthesis, ionic content, phytohormones, elements, and alkaloids in two ecotypes of Festuca sinensis under two soil water conditions.

The results and discussion were showed clearly, good works. However, the authors have previously reported its similar research; Effect of an Epichloë endophyte on adaptability to water stress in Festuca sinensis (Fungal Ecology, 2017, 30, 39-47, https://doi.org/10.1016/j.funeco.2017.08.005). Due to the lack of novelty, this article is not acceptable.

Author Response

Thank you for your agreement and summary of our manuscript. The previous Wang et al. (2017) research “Effect of an Epichloë endophyte on adaptability to water stress in Festuca sinensis” mainly investigated and evaluated the growth parameters such as plant height, number of tillers, total biomass and relative water content. In addition, Wang et al. are members of our research team. Based on these results, our study investigated and evaluated the photosynthesis, ionic content, phytohormones and alkaloids which related to drought stress to further study the specific physiology mechanism of enhancing the resistance of F. sinenses to drought stress. Our study is an important complement to the previous work and those paremeters was investigated to F. sinensis for the first time. Therefore, our study may be interesting and innovative.

Reviewer 2 Report

Comments to the manuscript plants-1311676 “Effects of Epichloë sinensis endophyte and host ecotype on physiology of Festuca sinensis under different soil moisture conditions”.

The manuscript is the report of an experiment of comparison between seedlings of two ecotypes of Festuca sinensis hosting the endophyte Epichloë sinensis (E+) and free of the infection (E-) under different watering regime (35% and 65% relative saturation moisture content). Many results are presented regarding plant biomass production, chlorophyll content, photosynthesis activity, ABA/IAA/CTK/GA content, potassium and calcium content, C/N/P content and relative ratios, demonstrating the influence of the endophyte presence as a stimulus to higher biological activity and best resistance to water stress. The manuscript represents a good contribute to the field of study and may be suitable for publication. However, I have two suggestions to authors in order to improve the article quality.

First of all, I want to ask to authors to associate to the data presentation a mean separation test like the Duncan’s Multiple Range Test in the Figures. Labelling the single histogram with letters may allow to understand the effective significant differences.

The second suggestion regards the results concerning the alkaloids that have been presented only for the E+ plants. I suggest to remove the Figure 10 to the Supplementary material and to be careful in the discussion where you speculate on the influence of the endophyte on the alkaloids concentration and the relative functions. The sentences reported in the lines 12-13, 27-29, 35, 67-71, 449-461, 478-482, and 554-557 are not appropriate or not justified by the experiment design and results presented.

Author Response

Thank you for your agreement and suggestions.

  1. The independent sample T-test allows us to understand the effect of ecotype and relative soil moisture condition more clearly, which is very important in our experiment. Your comments made us realize that readers may analyze our experimental results from different perspectives. After we have carefully studied your comments, repeated-measures ANOVA with Fisher's least significant differences (LSD) test was added and single histogram with letters were labelled to allow the readers understand the effective significant differences better in revised.

  1. After we have carefully studied your comments, we removed the Figure 10 and the sentences reported in the lines 12-13, 27-29, 35, 67-71, 478-482 and 554-557 in revised. We rewrited the relevant content in the lines 449-461 and reserved the possible role of alkaloids in drought resistance in the discussion of the article for readers to understand our research comprehensively.

Reviewer 3 Report

Dear Authors,

I have found your article very interesting and also in the field of my interest.

However, I would like to give you some small remarks and corrections.

First, consider methodology. In 4.2. Experimental Design row # 508 you mention that plants (i.e. seedlings) selected for the experiment were "of similar size"... In my opinion, the initial selection of 'similar' plants reduces the effect of natural variation between genotypes (one seed-one plant-one genotype) of plants of wind-pollinated, heterozygotic species. This could further influence differences in all observed and measured traits.

Second, considering statistics: Why didn't you apply typical ANOVA ?
You will get info about the main effect of factors as endophyte status and grass ecotype as well as their interaction on examined traits. 

Third, consider figures. As I understand 'whiskers' above boxes represent standard errors of means? Why didn't you place such an explanation in this place i.e. in the figure caption?

Sincerely

Author Response

Thank you for your recognition and high evaluation of my manuscript.

  1. Our purpose focused on the effects of endophyte, host ecotype and water condition. It is necessary to select plants of similar size in our experiment to reduce the errors caused by other factors before the starting of water stress. In addition, our team always select seedlings of similar size in experiments. For examples:

The studies of “Chen et al. (2014) - An asexual Epichloë endophyte modifies the nutrient stoichiometry of wild barley (Hordeum brevisubulatum) under salt stress.  Plant and Soil 387:153–165”, “Xia et al. (2018) - Effect of Epichloë gansuensis endophyte and transgenerational effects on the water use efficiency, nutrient and biomass accumulation of Achnatherum inebrians under soil water deficit. Plant and Soil. 424:555–571” and “Wang et al. (2017) - Effect of an Epichloë endophyte on adaptability to water stress in Festuca sinensis. Fungal Ecology 30:39-47” all used methods similar to ours.

  1. The main purpose of our experiment is to analyze the effects of endophyte, host ecotype and relative soil moisture condition in drought resistance of F. sinensis. The independent sample T-test allows us to understand the effect of host ecotype and relative soil moisture condition more clearly. After we have carefully studied your comments, we added the analysis of typical ANOVA in Figures and supplementary material of revision manuscript to get info about the main effect of factors as endophyte standard and grass ecotypes as well as their interaction better.

  1. The ‘whiskers’ above boxes represent standard errors of means. We added the information in the figure captions of all manuscript. Thank you for your suggestion again.

Round 2

Reviewer 1 Report

Dear Authors,

This article has been revised.

I accepted it.